# Validation and Clinical Application of the Japanese Version of the Patient-Reported Experience Measures for Intermediate Care Services: A Cross-Sectional Study

**DOI:** 10.3390/healthcare12070743

**Published:** 2024-03-29

**Authors:** Yuko Goto, Hisayuki Miura

**Affiliations:** Department of Home Care and Regional Liaison Promotion, Hospital, National Center for Geriatrics and Gerontology, Obu 474-8511, Aichi, Japan; hmiura@ncgg.go.jp

**Keywords:** intermediate care, community-based integrated care system, regional comprehensive care system, patient-centered care, community-based care ward, community comprehensive care ward, Kaifukuki rehabilitation ward, convalescence rehabilitation ward, patient-reported experience measure

## Abstract

Older adult patients account for 70% of all hospitalized patients in Japan, and intermediate care based on patient-centered care (PCC) that ensures continuity and quality of care at the interface between home services and acute care services and restores patient’s independence and confidence is necessary for them to continue living independently. At present, no concept of intermediate care is established in Japan, and the implementation of PCC has been delayed. Thus, in this study, a Japanese version of the intermediate care evaluation index (patient-reported experience measure (PREM)) was created on the basis of the original PREM developed in the UK, and data in wards with intermediate care functions in Japan were collected to confirm internal consistency and validity from 2020 to 2022. The Japanese version of PREM was found to have a factor structure with two potential factors. Given the clear correlation with the shared decision-making evaluation index, which is the pinnacle of PCC, the theoretical validity of the Japanese version of PREM, which is based on PCC as a theoretical basis, was confirmed.

## 1. Introduction

Globally, the proportion of older adults continues to increase, mainly in developed countries [1,2,3], and medical care systems have adapted to changes in population structure [4,5]. In Japan, the older adult population accounts for 30% of the total population [6], and considering that older adults account for 70% of all hospitalized patients [7], changing the healthcare system to meet the healthcare needs of the population will allow for more efficient use of healthcare resources [8]. Older adult patients often suffer from multiple chronic diseases such as hypertension and diabetes, and they often have disabilities specific to older adults, such as restrictions on daily living activities and frailty [9,10,11]. Therefore, providing highly individualized support integrating medical services, nursing care services, and daily living support for older adult patients is desirable [12].

In the 21st century, society required the enhancement of acute care to cure acute diseases, and since around 2000, the enhancement of primary care to support illnesses in daily life has been promoted. In recent years, the idea of “intermediate care”, which connects acute care and primary care, has spread in various countries in relation to various policies and methods.

Intermediate care is spreading mainly in Europe, and various policy definitions and concepts of intermediate care are being presented. In addition, efforts exhibited in intermediate care are advancing. The international definition of intermediate care that was announced in 2020 is as follows: “There was an agreement that intermediate care requires time-limited services that ensure continuity and quality of care, promote recovery, and restore independence and confidence at the interface between home and acute services, with transitional care representing a subset of intermediate care. Models are best delivered by an interdisciplinary team within an integrated health and social care system where a single contact point optimizes service access, communication, and coordination” [13].

In the UK, which pioneered intermediate care, intermediate care is defined as follows: “Intermediate care is a range of integrated services to promote faster recovery from rehabilitation, prevention of acute hospital admission, and admission to long-term residential care, support timely discharge from hospital, and maximize independent living” [14].

Moreover, a Cochrane review on nurse-led intermediate care published in 2007 indicated a potential decrease in early readmissions [15].

Considering that Japan is also a super-aging society and a society with a declining population [6], there is a strong demand for effective utilization of limited medical resources. Therefore, promoting short hospital stays and support to avoid inappropriate hospital admission/re-admission is important. Given that the relationship between unplanned readmissions and inadequate discharge care has been clarified, the possibility of improvement has been pointed out [16]. Pre-discharge interventions that are performed only within acute hospitals have various benefits that promote early discharge to avoid inappropriate hospital admission/re-admission. However, the importance of a system that provides integrated services centered on the individuality of patients has been pointed out during the transition period that spans between acute care services and at-home services [17].

Methods for providing intermediate care in recent years are classified into four categories [18]. The first category is intermediate care carried out only in hospitals during the transition period of care. The second category is intermediate care for transition care interventions performed at the time of discharge and within 30 days of discharge. The third category is intermediate care provided at home. Finally, the fourth category is intermediate care provided at local hospitals and facilities after receiving acute care. These categories of intermediate care indicate the difficulty of providing complex care in accordance with the individuality of patients, but such categories lead to positive outcomes among older adult patients [18].

In Japan, a long-term care insurance system was introduced in 2000 as the number of older adult patients increased [19]. Although the concept of intermediate care has not yet been defined in Japan, a “recovery-stage rehabilitation ward” (Kaifukuki Rehabilitation Ward: KRW) where older patients are provided care to stabilize their condition and encourage a lifestyle that maintains their physical function after they receive acute medical care was established in 2000 [20]. Furthermore, in 2014, a “community-based comprehensive care ward” (community-based care ward (CBCW)) was established and operated with enhanced preparation for life after discharge so that people can be safely discharged to a place where they are accustomed to living [21]. These forms of intermediate care are provided at hospitals in Japan.

Intermediate care is a new concept in Japan, and no evaluation indicators have been established to measure the quality of intermediate care performed by hospitals, such as KRW or CBCW, which can be used in Japanese. Therefore, the need to create a Japanese version of the patient-reported experience measure (PREM) developed in the UK, which is the pioneering policy development for intermediate care, has increased.

According to the international definition, intermediate care is care provided by an interdisciplinary team [13]. The operating regulations in Japanese acute care hospitals strictly stipulate the number of nurses assigned to a single ward, and the providers of intermediate care in Japanese acute care hospitals are primarily nurses.

Intermediate care is patient-centered integrated care suitable for various chronic diseases and physical conditions specific to older adults [13]. In addition, PREM has been developed using patient-centered care (PCC) as the basic theory [22], and patients’ intermediate care experience has been measured [22,23]. Quality healthcare must be patient-centered [24], and measuring patient experience leads to an assessment of whether it is functioning to meet individual patient needs [25]. Moreover, good patient care experience reports lead to an improvement in treatment outcomes [26,27]. Efforts are underway in Japan to improve the quality of PCC by measuring patient experience, mainly in the primary care area [28,29], and given the absence of evaluation indicators related to intermediate care functions, a Japanese version of PREM was created; its reliability and validity were the objectives of this study. Thus, this study aimed to develop and validate the novel Japanese version of PREM to improve the quality of intermediate care provided by Japanese hospitals.

## 2. Materials and Methods

### 2.1. Instruments

In this study, we used two types of evaluation indices: the Japanese version of PREM (bed base) and SDM-C Japanese.

#### 2.1.1. Japanese Version of PREM (Bed Base) (Appendix A)

PREM was developed using the Picker Institute’s Eight Principles of PCC as the basic theory, and a patient assessment composed of 15 items was used to measure the quality of intermediate care [22].

The eight principles of PCC are as follows: respect for patients’ values, preferences, and expressed needs; coordination and integration of care; information, communication, and education; physical comfort; emotional support and alleviation of fear and anxiety; involvement of family and friends; continuity and transition; and access to care [30].

PREM has been developed in two versions: bed-based (e.g., community hospitals) and home-based (e.g., hospital at home). The original PREM was developed in collaboration with intermediate care professionals, intermediate care practitioners, patient group representatives, and the Picker Institute [22].

The original version of PREM was developed by the UK’s National Health Service (NHS) and used for intermediate care evaluations in the UK [22]. In creating the Japanese version of PREM, approval to create the Japanese version was obtained from the British NHS’s Dr. Elizabeth Teal, who is the representative of the PREM original version developer team from 2017.

In 2018–2019, in accordance with the guidelines for cross-cultural adaptation [31], two Japanese nurses with experience in nursing work in English-speaking countries separately translated the original version of PREM into Japanese. The two Japanese nurses, together with one researcher with knowledge of intermediate care, one researcher with development research experience on translation scales, one manager each from KRW and CBCW—which are intermediate care companies in Japan—and one representative of the patient association, prepared a Japanese translation of one PREM adapted to the real situation in Japan.

Furthermore, one Japanese nurse with experience in nursing work in an English-speaking country performed a reverse translation from Japanese to English and confirmed that no difference was found in the content in 2018–2019.

#### 2.1.2. SDM-C Japanese

SDM is a patient-centered decision support method. While various SDM scales have been developed around the world, the SDM-C Japanese version is the only SDM scale developed in Japanese and confirmed to be reliable and valid for use in decision support for care in Japan.

SDM-C Japanese is a patient-reported experience measure with a one-factor structure that measures shared decision-making (SDM) [32], which is the pinnacle of PCC. SDM-C Japanese has been developed in Japanese, and its internal consistency and conceptual validity have been confirmed in Japanese clinical practice.

SDM-C Japanese was measured with the Japanese version of PREM simultaneously to verify the concurrent validity of the Japanese version of PREM, which measures PCC [33].

Factor 1 is composed of nine items, and all questions are answered using a six-level Likert scale, from “completely disagree” to “completely agree”.

#### 2.1.3. Demographic Data

Data on respondents’ frameworks (patients themselves or patients’ families), age, gender, and educational attainment were collected.

### 2.2. Study Design

This cross-sectional study used a questionnaire.

### 2.3. Participants and Setting

This study targeted patients admitted to one acute hospital of KRW and CBCW specializing in geriatrics research between December 2020 and August 2022.

The instruments were distributed to the patients by ward nurses or office workers just before the patients were discharged from the hospital. After being discharged from the hospital, the patients answered anonymously of their own free will and mailed the completed questionnaire using an anonymous envelope.

### 2.4. Ethical Considerations

This study was conducted after a strict conflict of interest and research ethics review by the National Center for Geriatrics and Gerontology. Approval (approval code no. 1434, 14 September 2020) was obtained from the National Center for Geriatrics and Gerontology. 

### 2.5. Statistical Analyses

For those with missing values of 30% or less of the total in the Japanese versions of PREM and SDM-C Japanese, statistical analysis was performed by substituting the median value of the relevant item. For demographic data, missing values were treated as “no response”.

The Japanese version of PREM and SDM-C Japanese summarized the response rates and response results.

Everything other than the Japanese version of PREM item 6 was scored as “present”, or the phenomenon that occurred was scored with “absent” and “I don’t know” set at 0. But the Japanese version of PREM item 6 was answered on a two-level Likert scale, and considering that “present” is a negative phenomenon, “absent” was set to 1 and “present” was 0. 

The statistical analysis method verified the validity and reliability of the Japanese version of PREM using the consensus-based standards for the selection of health measurement instruments (COSMIN) [34] as a reference.

The sample size was 100 or more, referring to COSMIN.

For statistical analysis, the Japanese version of PREM was analyzed by multiplying by 20/3, so the total number of answers to all 15 questions was 100.

SDM-C Japanese consists of nine questions, and all responses are based on a six-level Likert scale. “Completely disagree” was replaced with 0 and “Completely agree” with 5, and they were analyzed by multiplying the total responses by 20/9 to convert them to 100.

The Japanese version of PREM summarizes the median value, average value, minimum value, maximum value, interquartile range, and standard deviation.

As the factor structure of the original PREM (15 items) is a one-factor structure [23], the weight of the responses was distributed so that each item was 1 and the sum of 15 questions was 15 in the Japanese version of PREM. The responses were Likert responses, and 2–5 options were set depending on the item.

IBM SPSS Statistics 29 and IBM SPSS Amos Graphics 29 (IBM Corp., Armonk, NY, USA) were used as analysis software.

#### 2.5.1. Construct Validity

First, Kaiser–Meyer–Olkin (KMO) sampling validity was measured, and sampling was determined to be appropriate with KMO ≤ 0.6. Next, exploratory factor analysis was performed, and the number of potential factors was searched.

Then, as a confirmatory factor analysis, the conceptual structure was confirmed by structural equation modeling (SEM). The suitability of SEM was determined as highly compatible in estimating a Chi-square value (*p* > 0.05), goodness of fit index (GFI) ≥0.85, adjusted goodness of fit index (AGFI) ≥0.85, root mean square error of approximation (RMSEA) ≤ 0.05, and adaptive fit index (CFI) ≥0.95.

#### 2.5.2. Concurrent Validity

In verifying the concurrent validity of the Japanese version of PREM, the concurrent validity was confirmed by calculating Spearman’s ranking correlation coefficient from the total responses of the Japanese version of PREM and the total responses of SDM-C Japanese.

The correlation coefficient was interpreted as a weak correlation when r = 0.2 to 0.4 (*p* < 0.05), a moderate correlation when r = 0.4 to 0.7 (*p* < 0.05), and a strong correlation when r ≥ 0.8 (*p* < 0.05).

#### 2.5.3. Internal Consistency

In verifying the reliability of the Japanese version of PREM, the Cronbach α coefficient was calculated, and internal consistency was confirmed; α > 0.75 was interpreted as internally consistent.

## 3. Results

Data were provided from 136 patients, and the data from 116 patients (85.3%) whose missing responses to the Japanese version of PREM and SDM-C Japanese were 30% or less for each questionnaire were used as the analysis set.

### 3.1. Participants’ Characteristics

Respondent data were obtained from 63 patients (54.3%) discharged from KRW and 53 (45.7%) patients discharged from CBCW.

The respondents consisted of 60 patients (51.7%), 53 family members (45.7%), and no response for 3 (2.6%) patients.

The largest number of patients was in their 80s (*n* = 38, 32.8%), followed by 70s (*n* = 35, 30.2%).

A total of 71 patients were women (61.2%), 44 were men (37.9%), and 1 provided no response (0.9%).

The final level of education completed for most patients was high school (*n* = 56, 48.3%), followed by junior high school (*n* = 23, 19.8%, Table 1).

### 3.2. Response Results for the Japanese Version of PREM

The response results for the Japanese version of PREM are summarized in Table 2.

### 3.3. SDM-C Japanese Response Results

The response results for SDM-C Japanese are shown in Table 3.

### 3.4. Descriptive Statistics for the Japanese Version of PREM

The response results from the Japanese version of PREM were replaced by 100, and the descriptive statistics are summarized in Table 4.

### 3.5. Construct Validity

KMO was 0.793, thereby indicating sampling validity.

When factor analysis was performed using the main factor method, four factors with eigenvalues of 1 or more were extracted, but the cumulative contribution rate of the four factors did not exceed 50%. Therefore, assuming a 1-factor structure from the scree plot results (Figure 1), the number of factors was fixed again to 1, and factor analysis was performed using the main factor method promax rotation. Moreover, the commonality after factor extraction was PREM item 1 = 0.026 and PREM item 6 = 0.005.

Therefore, when the number of factors was fixed to 2, and factor analysis was performed using the main factor method, the commonality after factor extraction was PREM item 1 = 0.030 and PREM item 6 = 0.036. Based on the structural matrix, factor loadings were confirmed for PREM items 1 and 6 only for factor 1, and load amounts for 13 PREM items other than PREM items 1 and 6 were confirmed for two factors. Therefore, the Japanese version of PREM is a structure with two potential factors, and PREM items 2–5 and 7–15 are structures with factor loads for the two factors (Table 5).

Factors involved in the entirety of PREM are called “Japanese PREM”, and factors other than PREM 1 and 6 were termed “patient centeredness”.

When a hierarchical factor analysis model with a two-factor structure was constructed for confirmatory factor analysis using SEM, the application of the model was poor, as a hierarchical factor analysis model has a two-factor structure, with Chi-square = 218.539 (*p* = 0.000), GFI = 0.794, AGFI = 0.723, RMSEA = 0.113, and CFI = 0.714 (Figure 2).

Therefore, we constructed a model based on a residual correlation (Figure 3). The factor load for PREM item 1 was 0.32, and the factor load for PREM item 6 was 0.20. The Chi-square fit of the model was 81.148 (*p* = 0.322), with GFI = 0.916, AGFI = 0.868, RMSEA = 0.024, and CFI = 0.989, making it a model of good fit.

### 3.6. Concurrent Validity

Spearman’s ranking correlation coefficient was calculated from the overall score of the Japanese version of PREM with 15 items and the overall score of the SDM-C Japanese with nine items. The correlation coefficient was 0.541 (*p* < 0.001), confirming a moderate correlation.

### 3.7. Internal Consistency

Cronbach’s optimal alpha was calculated to verify the internal consistency in the Japanese version of PREM, and given that α = 0.804, sufficient internal consistency was confirmed.

## 4. Discussion

In this study, we created a bed-based intermediate care evaluation index that can confirm internal consistency and validity in Japanese clinical practice.

### 4.1. Descriptive Statistics for the Japanese Version of PREM

The policy concept for intermediate care has not been defined in Japan, but in this study, a Japanese version of PREM (bed base) was investigated for patients hospitalized in wards (KRW and CBCW) with intermediate care functions within a national center specializing in medical care for older adults.

Approximately half of the respondents were patients themselves, and the other half were their family, which is understandable because the age of the patients surveyed was mainly 70s and 80s.

In addition, in Japan, 70% of hospitalized patients are older adults [7]. Thus, the respondents to this survey are thought to be in the general age group of patients in Japan.

Of the respondents, 60% were women because the number of older adult women is increasing worldwide [35].

The fact that the final level of education of the respondents was high school is a consistent result, considering that women tend to have a lower level of education compared with men in Japan, and more female respondents were included in this study [36].

Based on the current survey results, the factor loadings for items 1 and 6 of the Japanese version of the one-factor first-order factor structure model of PREM were insufficient.

The Japanese version of PREM item 1 “The length of time I had to wait for the service to start was reasonable” is an interpretation of “patient centeredness” where the factor load is low, and hospitals and facilities primarily perform bed control in Japan. This result could reflect the current situation in Japan; thus, further research is necessary.

The Japanese version of PREM item 6 “I felt threatened or made to feel threatened by other patients or visitors during my stay in this service” is an interpretation of “patient centeredness” where the factor load is low, and a safe medical treatment environment from the patient’s point of view was not prepared sufficiently. The low factor loadings for items 1 and 6 imply possible concerns with the understanding or significance of these elements within the Japanese framework. Additional inquiry or improvement may be necessary.

Although the amount of load on “patient centeredness” was low for PREM item 13 “Staff participation with me whether additional equipment or adaptations were required to support me living at home”, a strong error correlation with other PREM items was confirmed. Therefore, staff participation mutually affects other PREM items and contributes to “patient centeredness”.

PREM item 13 is a question about discussions with care staff regarding preparations for community care.

Prior research on PCC in primary care in Japan has revealed that hospital-based practices are less community-oriented than community-based office practices [37]. In current Japanese intermediate care wards, patient-centered adjustments are not being made with an eye on how patients live in the community after being discharged from the hospital; thus, further research is necessary.

In the original English version of PREM (bed base), eight items of PREM, namely, 2, 4, 7, 8, 10, 11, 12, and 15, showed a moderate scale in accordance with Mokken analysis [22]. In addition, PREM items 2 and 14 were excluded from the Italian PREM (bed base) [23].

Regarding the development of the translated scale, issues with cultural adaptation in each region have been pointed out [38,39]. Issues with implementing intermediate care centered on PCC in each country have also been determined; thus, efforts aimed at the social implementation of PCC based on each country and culture are necessary.

Here, for the translation, multiple discussions were held with healthcare professionals who are fluent in English, researchers with scale development experience, researchers with knowledge of intermediate care, and multiple ward managers with intermediate care functions in Japan, and expressions suitable for intermediate care in Japan were prepared. The 13 PREM items other than PREM items 1 and 6 converge to factor 1, and such items may have been prepared with an appropriate expression that can measure the quality of intermediate care for Japanese bed bases.

The Japanese version of PREM, which has PCC as the basic theory, and SDM-C Japanese, which is an evaluation index of SDM that is regarded as the pinnacle of PCC, showed a clear correlation. The Japanese version of PREM created in this study effectively measures PCC. Therefore, the Japanese version of PREM is an evaluation index that can measure bed-based intermediate care that can be used in Japanese based on PCC with confirmed conceptual validity.

### 4.2. Significance of Bed-Based PREM Development as the PCC Basic Theory in Japan

This study was the first to create a bed-based intermediate care evaluation index in Japanese, which can be used in Japan based on PCC as a theoretical basis. The concept of intermediate care has not been established in Japan to date. And in Japan, PCC education is not sufficiently carried out in basic education for doctors, etc. [40].

Japan has adopted a free-access medical care system for patients, and patients’ medical satisfaction has been reported to be high in accordance with surveys within Japan [41]. On the other hand, there are public opinions voicing dissatisfaction with the lack of public representation in making health policies, and patient and public healthcare systems must include patient and public involvement, such as patient-centered perspectives, which is being promoted globally [41]. Therefore, Japan is in a situation where promoting changes in the medical system that incorporates the PCC viewpoint and education of medical personnel is necessary.

In addition, in Japan, the older adult population accounts for 30% of the total population [6], and many older adult patients are in need of intermediate care [13], which consists of patient-centered integrated care suitable for various chronic diseases and physical conditions specific to older adults.

The Institute of Medicine points out that PCC is important for high-quality medical care [24], and the Japanese version of PREM, which is based on the PCC created in this study, may be of great importance to future Japanese medical care.

In Asia, including Japan, there is a culture that values family-centeredness rather than patient-centeredness [42,43]. Prior research in East Asia has revealed that the more family members there are, the lower the SDM between clinicians and patients [44], and even in Japan, a clinical ethics mindset advocates the importance of decision support centered on the patient’s family [45].

According to a previous study comparing outpatient PCC in individualistic Australia and family-oriented Japan, patients in Australia compared to Japan were asked if they would prefer to have a friend or family member present during treatment consultations. [46]. In Japan, it is “natural” for family members to attend treatment explanations, and patients rarely confirm consent. In Japan, patients are not present, and it is not uncommon for doctors to explain their medical conditions only to family members. In the Japanese medical field, the PCC way of thinking is gradually being introduced, but many issues have arisen in actual practice [47,48].

PCC has yet to be implemented in Japan, and no concept of intermediate care has been proposed at present. However, efforts can proceed toward implementation of intermediate care based on PCC using the Japanese version of PREM in Japan hereafter.

Furthermore, future research must contribute to intermediate care education based on PCC as a theoretical basis for healthcare workers.

### 4.3. Strengths and Limitations of This Study

This study was conducted in a ward with two intermediate care functions in one Japanese National Center facility that specializes in medical care for older adults. This can restrict the applicability of the results. Thus, surveys targeting facilities with more intermediate care functions must be conducted in the future.

Subsequent research should incorporate a more extensive array of individuals with vested interests, such as patients, families, healthcare professionals, and policymakers, in the creation and assessment of intermediate care evaluation indices.

This survey was also conducted in the context of the 2020-to-2022 COVID-19 pandemic, and communication between care providers and patients may have been different. Given the proportion of older adult patients is high in Japan, medical institutions are still dealing with severe infections, and they have set communication restrictions. Therefore, conducting research on bed-based intermediate care is necessary, including medical institutions that accept patients of various age groups.

## 5. Conclusions

In this study, we created a bed-based intermediate care evaluation index that can confirm internal consistency and validity in Japanese clinical practice. The concept of intermediate care has not been developed in Japan, and PCC has yet to be implemented in the real world. However, efforts aimed at enhancing intermediate care and improving the quality of medical care will be promoted using the Japanese version of PREM created in this study. Research contributing to intermediate care education based on PCC as a theoretical basis for healthcare workers is also necessary.

## Figures and Tables

**Figure 1 healthcare-12-00743-f001:**
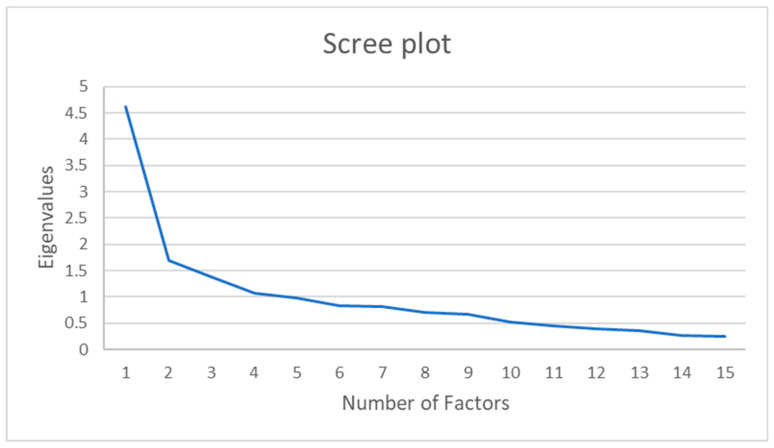
Scree plot based on exploratory factor analysis of the Japanese version of PREM.

**Figure 2 healthcare-12-00743-f002:**
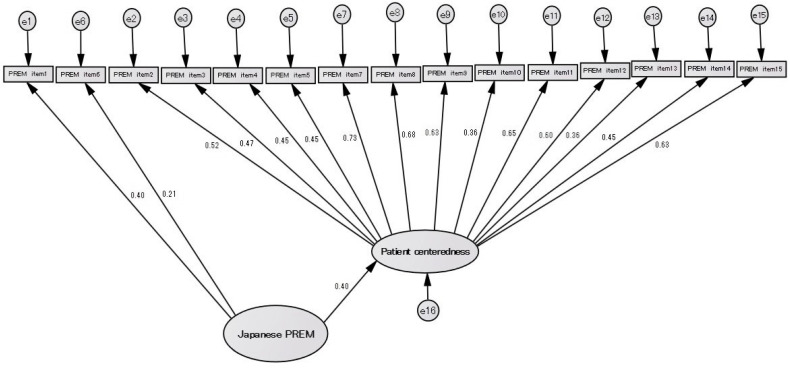
Hierarchical factorial analysis model with a two-factor structure with no residual correlation.

**Figure 3 healthcare-12-00743-f003:**
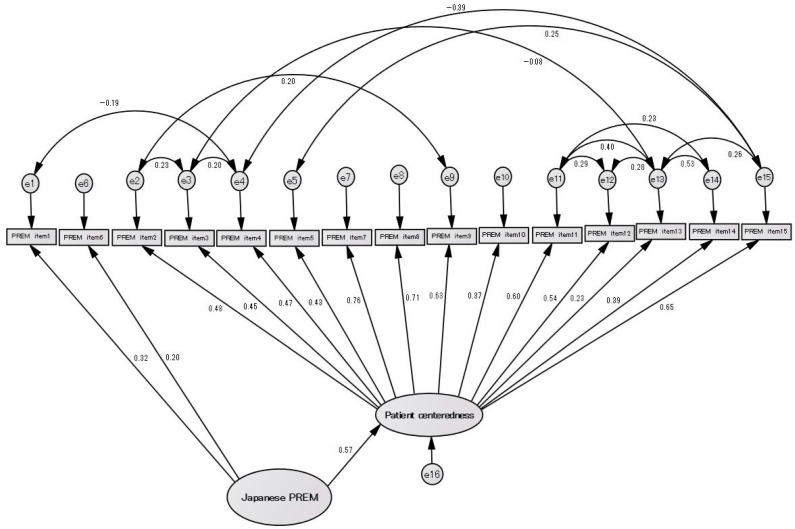
Hierarchical factor analysis model with a two-factor structure with residual correlation.

**Table 1 healthcare-12-00743-t001:** Participants’ characteristics (*n* = 116).

Category	Response Choices	*n*	%
Respondent	Patient	60	51.7
	Family of the patient	53	45.7
	No response	3	2.6
Patient age group	40 years or less	3	2.6
	50s	18	15.5
	60s	12	10.3
	70s	35	30.2
	80s	38	32.8
	90 and over	9	7.8
	No response	1	0.9
Patient’s gender	Female	71	61.2
	Male	44	37.9
	No response	1	0.9
Patient’s final level of education completed	Junior high school	23	19.8
	Senior high school	56	48.3
	National Colleges of Technology	6	5.2
	University/Graduate School	21	18.1
	No response	10	8.6

**Table 2 healthcare-12-00743-t002:** Response results for the Japanese version of PREM items (*n* = 116).

PREM Items [22]	Response Choices	*n*	%
PREM item 1 The length of time I had to wait for the service to start was reasonable	Yes	109	94.0
	No	7	6.0
PREM item 2 The staff that cared for me in this service had been given all the necessary information about my condition or illness from the person who referred me	Yes	80	69.0
	No/I don’t know	36	31.0
PREM item 3 I was aware of what we were aiming to achieve, e.g., to be mobile at home, to be independent at home, to be able to go out shopping, to understand my health better	Yes	88	75.9
	No/I don’t know	28	24.1
PREM item 4 I was involved in setting these aims	I always thought of it	62	53.4
	I sometimes thought of it	44	37.9
	I do not think of it	10	8.6
PREM item 5 The room or ward I was in was clean	It was very clean	51	44.0
	It was clean	62	53.4
	It was not clean	3	2.6
PREM item 6 I felt threatened or made to feel uncomfortable by other patients or visitors during my stay in this service	No	97	83.6
	Yes	19	16.4
PREM item 7 When I had important questions to ask the staff they were answered well enough	Yes	63	54.3
	To some extent	43	37.1
	I didn’t need to ask	7	6.0
	They didn’t answer	3	2.6
PREM item 8 I had confidence and trust in the staff treating or supporting me	I could trust them	82	70.7
	I could trust them to some extent	33	28.4
	I could not trust them	1	0.9
PREM item 9 I was involved as much as I wanted to be in decisions about my care and therapy	Yes	92	79.3
	No	24	20.7
PREM item 10 I was involved in decisions about when I would go home	Adequately involved	59	50.9
	Somewhat involved	37	31.9
	No need for me to be involved	9	7.8
	Not involved	11	9.5
PREM item 11 Staff took account of my family or home situation when planning going home	Well thought of	66	56.9
	To some extent	33	28.4
	Was not necessary	8	6.9
	Not thought of Don’t know	9	7.8
PREM item 12 Staff gave my family or someone close to me all the information they needed to help care for me	Ample information given	55	47.4
	Information given to some extent	49	42.2
	I did not want the information given/ Was not necessary	7	6.0
	No information given	5	4.3
PREM item 13 Staff discussed with me whether additional equipment or adaptations were required to support me living at home	Good discussion	70	60.3
	No need to discuss	37	31.9
	I wanted to discuss Could not discuss	9	7.8
PREM item 14 Staff discussed with me whether I needed any further health or social care services after this service stopped. (e.g., services from a GP, physiotherapist or community nurse, or assistance from social services or the voluntary sector)	Good discussion	71	61.2
	No need to discuss	33	28.4
	I wanted to discuss Could not discuss	12	10.3
PREM item 15 Overall, I felt I was treated with respect and dignity while I was receiving my care from this service	Always	83	71.6
	Sometimes	30	25.9
	None	3	2.6

**Table 3 healthcare-12-00743-t003:** SDM-C Japanese response results (*n* = 116).

SDM Items	Response Choices	*n*	%
SDM item 1 My care staff made clear to me that care decision needs to be made	Strongly agree	37	31.9
	Somewhat agree	42	36.2
	Possibly agree	24	20.7
	Somewhat disagree	5	4.3
	Disagree	5	4.3
	Strongly disagree	3	2.6
SDM item 2 My care staff wanted to know exactly how I want to be involved in making the decision	Strongly agree	35	30.2
	Somewhat agree	42	36.2
	Possibly agree	21	18.1
	Somewhat disagree	13	11.2
	Disagree	1	0.9
	Strongly disagree	4	3.4
SDM item 3 My care staff told me that there are different options for treating my condition	Strongly agree	35	30.2
	Somewhat agree	40	34.5
	Possibly agree	21	18.1
	Somewhat disagree	12	10.3
	Disagree	4	3.4
	Strongly disagree	4	3.4
SDM item 4 My care staff precisely explained the advantages and disadvantages of the care options	Strongly agree	31	26.7
	Somewhat agree	31	26.7
	Possibly agree	29	25.0
	Somewhat disagree	15	12.9
	Disagree	6	5.2
	Strongly disagree	4	3.4
SDM item 5 My care staff helped me understand all the information	Strongly agree	37	31.9
	Somewhat agree	35	30.2
	Possibly agree	28	24.1
	Somewhat disagree	10	8.6
	Disagree	3	2.6
	Strongly disagree	3	2.6
SDM item 6 My care staff asked me which care option I prefer	Strongly agree	37	31.9
	Somewhat agree	38	32.8
	Possibly agree	21	18.1
	Somewhat disagree	10	8.6
	Disagree	5	4.3
	Strongly disagree	5	4.3
SDM item 7 My care staff and I thoroughly weighed the different care options	Strongly agree	27	23.3
	Somewhat agree	43	37.1
	Possibly agree	23	19.8
	Somewhat disagree	12	10.3
	Disagree	5	4.3
	Strongly disagree	6	5.2
SDM item 8 My care staff and I selected a care option together	Strongly agree	31	26.7
	Somewhat agree	35	30.2
	Possibly agree	31	26.7
	Somewhat disagree	6	5.2
	Disagree	7	6.0
	Strongly disagree	6	5.2
SDM item 9 My care staff and I reached an agreement on how to proceed	Strongly agree	31	26.7
	Somewhat agree	40	34.5
	Possibly agree	23	19.8
	Somewhat disagree	10	8.6
	Disagree	7	6.0
	Strongly disagree	5	4.3

**Table 4 healthcare-12-00743-t004:** Descriptive statistics for the Japanese version of PREM.

PREM Items	Median	Minimum Value	Maximum Value	Dispersion	Mean	Standard Deviation
PREM item 1	6.67	0.00	6.67	2.542	6.26	1.594
PREM item 2	6.67	0.00	6.67	9.595	4.60	3.098
PREM item 3	6.67	0.00	6.67	8.209	5.06	2.865
PREM item 4	6.67	0.00	6.67	4.704	4.83	2.169
PREM item 5	3.34	0.00	6.67	3.298	4.71	1.816
PREM item 6	6.67	0.00	6.67	6.140	5.58	2.478
PREM item 7	6.67	0.00	6.67	2.589	5.41	1.608
PREM item 8	6.67	0.00	6.67	2.555	5.66	1.598
PREM item 9	6.67	0.00	6.67	7.356	5.29	2.712
PREM item 10	6.67	0.00	6.67	4.521	4.99	2.126
PREM item 11	6.67	0.00	6.67	4.131	5.22	2.033
PREM item 12	4.47	0.00	6.67	2.981	5.18	1.727
PREM item 13	6.67	0.00	6.67	4.534	5.09	2.129
PREM item 14	6.67	0.00	6.67	5.120	5.03	2.263
PREM item 15	6.67	0.00	6.67	2.979	5.63	1.726

**Table 5 healthcare-12-00743-t005:** Factor structure matrix based on factor analysis where 2 is fixed as the factor number.

	Factor 1	Factor 2
PREM item 7	0.735	0.325
PREM item 8	0.721	0.237
PREM item 9	0.643	0.268
PREM item 15	0.607	0.397
PREM item 2	0.537	0.262
PREM item 12	0.532	0.518
PREM item 3	0.530	0.142
PREM item 5	0.490	0.167
PREM item 4	0.433	0.317
PREM item 10	0.368	0.191
PREM item 1	0.171	
PREM item 6	0.115	
PREM item 13	0.195	0.921
PREM item 14	0.350	0.668
PREM item 11	0.577	0.626

## Data Availability

The data used to support the findings of this study are available from the corresponding author upon request.

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
