# Peer review of "Validation and Clinical Application of the Japanese Version of the Patient-Reported Experience Measures for Intermediate Care Services: A Cross-Sectional Study"

_healthcare, 2024, doi:10.3390/healthcare12070743_

Round 1
Reviewer 1 Report
Comments and Suggestions for Authors
Dear authors
I would like to thank you for giving me the opportunity to review the manuscript entitled “Validation and clinical application of the Japanese version of the patient-reported experience measures for intermediate care services: A cross-sectional study”. This manuscript is well-designed and -written. I think it can be considered for publication after revision. I have some comments that you can be considered:
- Can you give more information regarding nurses' role in intermediate care? you can use the following concept analysis article: https://onlinelibrary.wiley.com/doi/full/10.1002/nop2.2083
- Please provide more information about the original version of PREM
- Is the sample size sufficient? It seems that the number of samples should be more.
- Please put the original version and the Japanese version of PREM as supplementary files
Author Response
1. COMMENT
Dear authors
I would like to thank you for giving me the opportunity to review the manuscript entitled “Validation and clinical application of the Japanese version of the patient-reported experience measures for intermediate care services: A cross-sectional study”. This manuscript is well-designed and -written. I think it can be considered for publication after revision. I have some comments that you can be considered:
RESPONSE:
Thank you very much for your review.
We followed your comments and worked diligently to improve this article.
2. COMMENT
Can you give more information regarding nurses' role in intermediate care? you can use the following concept analysis article: https://onlinelibrary.wiley.com/doi/full/10.1002/nop2.2083
RESPONSE:
The purpose of this study was to develop a Japanese version of the intermediate care evaluation index for intermediate care in acute care hospitals. To this end, an international definition of intermediate care providers and information on intermediate care providers in Japanese acute care hospitals were added to the Introduction.
3. COMMENT
Please provide more information about the original version of PREM
RESPONSE:
The information about the original version of PREM has been added to "2.1.1 Japanese PREM (bed base)".
4. COMMENT
Is the sample size sufficient? It seems that the number of samples should be more.
RESPONSE:
A sample size of at least 100 samples was accumulated according to COSMIN guidelines, and the KMO sample validity analysis was 0.793, which was considered an adequate sample size for the analysis.
5. COMMENT
Please put the original version and the Japanese version of PREM as supplementary files
RESPONSE:
Since the original version is posted on the web page of the original PREM paper and has a problem of copyright, we could not put it. Therefore, we put only the Japanese version of the PREM as supplemental files.

Reviewer 2 Report
Comments and Suggestions for Authors
The authors presented a well designed validation study of the Japanese translation of the PREM. Generally speaking the paper is well written (my only concern is the potential low interest for international reader due to the nature of the paper but this has nothing to do with the quality of the manuscript), here ar my specific comments.
Introduction:
The introduction offers a nice overview of the background, encompassing the worldwide phenomenon of a global aging of the population, the particular circumstances in Japan, and the emerging notion of intermediate care. This contextualization establishes a solid basis for the presnted work.
Although the introduction discusses the importance of assessing intermediate care in Japan and developing a Japanese version of PREM, it would be advantageous to include a more explicit explanation of the unique contribution of the study. Providing a clear explanation of how the study addresses gaps in knowledge or technique would increase its significance.
Methods and results:
The methods are well explained and the results are given in a clear and concise manner.
Discussion:
The study indicates that items 1 and 6 of the Japanese version of PREM exhibit inadequate factor loadings. This implies possible concerns with the understanding or significance of these elements within the Japanese framework. Additional inquiry or improvement may be necessary.
The study recognizes that it was carried out in a particular environment within a solitary national center establishment, which restricts the applicability of the results. It indicates the necessity for more comprehensive studies that focus on facilities with a wider range of intermediate care functions.
Points for improvement:
The study recognizes the difficulties of adapting to different cultures while translating and implementing evaluation techniques such as PREM in various countries. Further investigation could explore these difficulties in greater depth and provide tactics for surmounting them in order to guarantee the accuracy and dependability of assessment tools.
Subsequent research should incorporate a more extensive array of individuals with vested interests, such as patients, families, healthcare professionals, and policymakers, in the creation and assessment of intermediate care evaluation indices. This would guarantee that the instruments are pertinent, thorough, and in harmony with the requirements and viewpoints of all stakeholders engaged in care provision.
Comments on the Quality of English LanguageEnglish is fine
Author Response
1. COMMENT
The authors presented a well designed validation study of the Japanese translation of the PREM. Generally speaking the paper is well written (my only concern is the potential low interest for international reader due to the nature of the paper but this has nothing to do with the quality of the manuscript), here ar my specific comments.
RESPONSE:
Since intermediate care and patient-centered care have not penetrated society in many areas, we would like to continue our research in order to generate interest in these areas in various regions.
2. COMMENT
Introduction:
The introduction offers a nice overview of the background, encompassing the worldwide phenomenon of a global aging of the population, the particular circumstances in Japan, and the emerging notion of intermediate care. This contextualization establishes a solid basis for the presnted work.
RESPONSE:
Thank you for your comment.
3. COMMENT
Although the introduction discusses the importance of assessing intermediate care in Japan and developing a Japanese version of PREM, it would be advantageous to include a more explicit explanation of the unique contribution of the study. Providing a clear explanation of how the study addresses gaps in knowledge or technique would increase its significance.
RESPONSE:
We added a sentence at the end of the introduction to clarify the strengths of this article.
4. COMMENT
Methods and results:
The methods are well explained and the results are given in a clear and concise manner.
RESPONSE:
Thank you for your comment.
5. COMMENT
Discussion:
The study indicates that items 1 and 6 of the Japanese version of PREM exhibit inadequate factor loadings. This implies possible concerns with the understanding or significance of these elements within the Japanese framework. Additional inquiry or improvement may be necessary.
RESPONSE:
Thank you for your comment. We added the contents of your comments to Discussion.
6. COMMENT
The study recognizes that it was carried out in a particular environment within a solitary national center establishment, which restricts the applicability of the results. It indicates the necessity for more comprehensive studies that focus on facilities with a wider range of intermediate care functions.
RESPONSE:
We corrected the sentence in “4.3 Strengths and limitations of this study” to clarify it.
7. COMMENT
Points for improvement:
The study recognizes the difficulties of adapting to different cultures while translating and implementing evaluation techniques such as PREM in various countries. Further investigation could explore these difficulties in greater depth and provide tactics for surmounting them in order to guarantee the accuracy and dependability of assessment tools.
RESPONSE:
Thank you for your comments. The adapting to different cultures is very challenging. We will try further to explore these difficulties in greater depth and provide tactics for surmounting them in order to guarantee the accuracy and dependability of assessment tools, as you mentioned.
8. COMMENT
Subsequent research should incorporate a more extensive array of individuals with vested interests, such as patients, families, healthcare professionals, and policymakers, in the creation and assessment of intermediate care evaluation indices. This would guarantee that the instruments are pertinent, thorough, and in harmony with the requirements and viewpoints of all stakeholders engaged in care provision.
RESPONSE:
Thank you for your comments. We added the contents of your comments to “4.3 Strengths and limitations of this study” to clarify it.

Reviewer 3 Report
Comments and Suggestions for Authors
Thank you for the opportunity to review the manuscript. The article presents the validation and clinical application of patient-reported experience measures in the context of intermediate care. Please find my suggestions.
Abstract
Please describe the date of the study.
Materials and Methods
Please describe the date of the different steps of Japanese version of PREM
Line 129: “2.1.2 SDM-C Japanese”: Please describe SDM
Line 134 “…shared decision-making..” Please write this description of SDM in line 130
Line 143: “This cross-sectional study used a questionnaire”. Did the participants respond to two questionnaires (Japanese version of PREM and SDM-C Japanese), right? Please describe this. The authors could write the average time needed to complete the questionnaires
Results
Table 1: “Patient’s physical sex” Please rewrite this.
Discussion
The authors could summarize the study with the positive results in the first paragraph of the discussion.
Author Response
1. COMMENT
Thank you for the opportunity to review the manuscript. The article presents the validation and clinical application of patient-reported experience measures in the context of intermediate care. Please find my suggestions.
RESPONSE:
Thank you very much for your review. We will improve this article according to your suggestion.
2. COMMENT
Abstract
Please describe the date of the study.
RESPONSE:
We had appended it.
3. COMMENT
Materials and Methods
Please describe the date of the different steps of Japanese version of PREM
RESPONSE:
We had appended it.
4. COMMENT
Line 129: “2.1.2 SDM-C Japanese”: Please describe SDM
RESPONSE:
We had appended it.
5. COMMENT
Line 134 “…shared decision-making..” Please write this description of SDM in line 130
RESPONSE:
We had moved it.
6. COMMENT
Line 143: “This cross-sectional study used a questionnaire”. Did the participants respond to two questionnaires (Japanese version of PREM and SDM-C Japanese), right? Please describe this. The authors could write the average time needed to complete the questionnaires
RESPONSE:
We had appended it.
7. COMMENT
Results
Table 1: “Patient’s physical sex” Please rewrite this.
RESPONSE:
We changed "physical gender" to "gender" in "2.1.3 Demographic data" in "2. Materials and Methods", and we changed "Patient's physical sex" to "Patient's gender" in Table 1.
8. COMMENT
Discussion
The authors could summarize the study with the positive results in the first paragraph of the discussion.
RESPONSE:
We added a sentence about the positive results to the first paragraph of the discussion.
